# Investigation of Eating Disorder Risk and Body Image Dissatisfaction among Female Competitive Cheerleaders

**DOI:** 10.3390/ijerph19042196

**Published:** 2022-02-15

**Authors:** Allison B. Smith, Jennifer L. Gay, Eva V. Monsma, Shawn M. Arent, Mark A. Sarzynski, Dawn M. Emerson, Toni M. Torres-McGehee

**Affiliations:** 1School of Kinesiology, University of Louisiana at Lafayette, Lafayette, LA 70507, USA; 2Department of Health Promotion & Behavior, University of Georgia, Athens, GA 30602, USA; jlgay@uga.edu; 3Department of Physical Education, University of South Carolina, Columbia, SC 29208, USA; eavadocz@mailbox.sc.edu; 4Department of Exercise Science, University of South Carolina, Columbia, SC 29208, USA; sarent@mailbox.sc.edu (S.M.A.); sarz@mailbox.sc.edu (M.A.S.); torresmc@mailbox.sc.edu (T.M.T.-M.); 5Department of Physical Therapy, Rehabilitation Science, and Athletic Training, University of Kansas Medical Center, Kansas City, KS 66160, USA; demerson@kumc.edu

**Keywords:** athletes, perceptions, meta-perceptions, aesthetic, pathogenic behaviors

## Abstract

Social agents associated with cheerleading environments are increasingly linked to body image dissatisfaction (BID) and eating disorders (ED). This study examined ED risk across team type, squad type, and position. An additional purpose determined BID in clothing type (daily clothing, midriff uniform, and full uniform), and meta-perceptions from the perspective of peers (MP peers), parents (MP parents), and coaches (MP coaches). Female cheerleaders (*n* = 268) completed an online survey which included demographics, the Eating Attitudes Test-26, and pathogenic behavior questions. Body image perceptions were assessed by using the Sex-Specific Figural Stimuli Silhouettes. Overall, 34.4% of cheerleaders (*n* = 268; mean age: 17.9 ± 2.7 years) exhibited an ED risk. Compared to All-Star cheerleaders, college cheerleaders demonstrated significant higher ED risk (*p* = 0.021), dieting subscale scores (*p* = 0.045), and laxative, diet pill, and diuretic use (*p* = 0.008). Co-ed teams compared to all-girl teams revealed higher means for the total EAT-26 (*p* = 0.018) and oral control subscale (*p* = 0.002). The BID in clothing type revealed that cheerleaders wanted to be the smallest in the midriff option (*p* < 0.0001, η2 = 0.332). The BID from meta-perception revealed that cheerleaders felt that their coaches wanted them to be the smallest (*p* < 0.001, η2 = 0.106). Cheerleaders are at risk for EDs and BID at any level. Regarding the midriff uniform, MP from the perspective of coaches showed the greatest difference between perceived and desired body image.

## 1. Introduction

Over the last three decades, cheerleading has grown in popularity as a competitive sport, with millions of participants across the United States of America [1,2]. Historically, cheerleading participation was predominately in the high school and collegiate settings. Cheerleaders’ primary responsibilities are to make appearances at large events (i.e., football and basketball games), to assist in crowd enthusiasm, and promote events during the academic year. While high school and college cheerleading is still present and popular today, a catalyst for the rapid growth of the sport was the creation of a new cheer category termed All-Star cheerleading, which encompasses competitors ranging from ages 5–18. All-Star cheerleading fosters a competitive arena where young athletes can showcase their abilities that merge dance, power tumbling, and partner stunting into a choreographed two-and-a-half minute routine. All-Star cheerleading teams are made up of members from a gym or club who compete multiple times throughout the year, and performances are evaluated and scored by a panel of judges. Comparatively, college cheerleaders continue to appear at large events and only have one annual competition that is evaluated and judged by a panel of judges. There are two squads: all-girl or co-ed (females and males), and cheerleading positions consist of flyer, base, and back spot. The flyer is typically thrown and completes acrobatic skills in the air [3]. The base and back spots, who are similar in overall size, are stronger due to the need for tossing, catching, and holding a flyer. Positions are the same across the various team and squad types, however the orientation of the positions may vary. Flyers are generally shorter, lighter, and have lower body mass indexes compared to other positions [4].

Females participating in aesthetic sports (e.g., gymnastics, figure skating, cheerleading) are at an increased risk for eating disorders (EDs) and pathogenic behaviors—binge eating, purging, self-induced vomiting, use of diet pills or laxatives, and fasting—compared to non-aesthetic sports and non-athletes [3,5,6,7,8,9,10,11,12,13,14]. Currently, several studies included the cheerleading population when examining ED risk [1,4,15,16,17]; however, none are focused on the younger populations, specifically under the 18-year-old threshold. Of these studies, the majority are outdated [1,15,16,17], used extremely low sample sizes (i.e., *n* = 1) [15], and included many other sports [15,16]. The risk of ED for cheerleaders ranged from 13–33%, with the flyer position being at the highest risk [1,3]. Currently, very few studies have investigated ED risk among cheerleaders specific to team type (college or All-Star) and squad type (all-girl or co-ed).

Risk factors for EDs include an athletic context that values a low body weight, small physique, subjective evaluation performance components, frequent weight cycling, and early sport specialization [16,18]. These factors likely predispose cheerleaders to body image dissatisfaction (BID), which is defined as a preoccupation with one’s own body [19,20]. Body image dissatisfaction has previously been documented as a risk factor for EDs within the cheerleading population, due to factors attributed to the objectification theory [9]. This theory is the act of an individual being treated as an object, rather than a person [21,22]. Within cheerleading, females are often objectified through the subjective judging practices which are based on how they appear in their competitive uniform compared to societal norms [23]. In addition to objectification, cheerleaders must also navigate meta-perceptions of social agents in the cheer environment. Meta-perceptions are one’s perceived perception on how their peers (MP peers), parents (MP parents), and coaches (MP coaches) see them, which is a contributing source of stress and affects the individual’s body image [4]. Cheerleading may force participants to strive for a level of perfection that is evaluated differently by MP peers, MP parents, and MP coaches. The variations in these body image perceptions may increase the amount of feedback or commentary that an individual cheerleader receives from their social agents, and may impact their mental health by increasing their overall stress levels, which has been linked to external behaviors such as constant body checking, body shaming, and anxiety [4,24]. Often, these behaviors can overpower internal commentary, such as feelings of hunger, which when ignored, may trigger behaviors in cheerleaders such as restricting food or self-induced vomiting to change their body weight and shape in hopes of increasing their performance and gaining positive feedback from their MP peers, MP parents, and MP coaches [4,24]. The behaviors linked to intrusive thoughts are especially important to screen, monitor, and circumvent as early as possible among youth cheerleaders becasue body image dissatisfaction being linked as a risk factor for EDs [9].

Additionally, uniform type contributes to 54–99% of collegiate athletes experiencing increased body-consciousness and increased negative self-consciousness [3,25]. In cheerleading, uniform types consists of a midriff uniform (the top reveals the midriff region—displaying skin around the stomach area) and full-length uniform (top is full-length—does not display any skin around the stomach region). Previous literature revealed that the largest predisposing ED risk factor among collegiate cheerleaders was midriff BID, accounting for 32% of the variance, followed by meta-perceptions from the perspective of MP parents, which accounted for 2% of the variance in eating disorder risk. Body image dissatisfaction from MP coaches along with social physique anxiety explained 5.6% and 19.1% of the variance in depression risk, respectively. In all analyses, greater body dissatisfaction, usually favoring being smaller, was associated with greater mental health risks [4]. Notably, BID symptomology begins in the teenage years, prior to the age of 18, pointing to closer examinations of youth cheerleaders who are enjoying expanding participation opportunities.

Given the youth cheerleader focus gap currently in the literature, this study had four aims. First, we examined the overall prevalence of ED risk, eating attitudes, and pathogenic behaviors of competitive cheerleaders aged 12 to 25 years old to encompass the all-star and college age specifications for cheerleading competitions. Second, ED risk, eating attitudes, and pathogenic behaviors across team type (All-Star or college), squad type (all-girl or coed), and position (flyer, base, back spot) were examined. Third, we examined perceived body image (PBI) perceptions vs. desired body image (DBI) of cheerleaders in various clothing types (daily clothing, midriff uniform, and full-length uniform). Finally, we examined cheerleaders’ meta-perception BID from the perspective of MP peers, MP parents, and MP coaches.

## 2. Materials and Methods

A total of 268 cheerleaders from across the United States participated in this cross-sectional study (mean age: 17.9 ± 2.7 years; All-Star: *n* = 134; college: *n* = 134). Participants were included if they were All-Star cheerleaders who were a member of a team that competed within the United States All-Star Federation (USASF) competition circuit. The USASF names divisions for competition based on two categories: skill level and age of competitors. Skill levels are broken down into levels ranging from 1–6, while age ranges are described as Tiny, Mini, Youth, Junior, and Senior. For this study, participants had to be competing within the skill level of 5, and within the Senior age division, which was delineated to 12–18 years old. Participants were excluded if they were not an active member of an All-Star or college team. This study was conducted in accordance with the Declaration of Helsinki, the study protocol was approved by the University of South Carolina’s Institutional Review Board (IRB-Pro00082027), and all participants consented prior to participation.

### 2.1. Instruments

#### 2.1.1. Demographic Information Survey

Basic demographic data were collected through an online survey, which included age, team type (All-Star or college), squad type (all-girl or co-ed), position (flyer, base, back spot), high school or college academic status, and years of experience in the sport. Participants self-reported their height, current weight, highest weight, lowest weight, and ideal weight.

#### 2.1.2. Eating Attitudes Test (EAT-26)

The EAT-26 was used to determine individuals at risk for EDs by using standardized measures of eating attitudes and behaviors [26]. The instrument is not a diagnostic tool, but is commonly used as a screening method to identify attitudes and behaviors that indicate a potential ED [26]. Three subscales include attitudes relative to dieting, bulimia, and food preoccupation/oral control. Five supplemental questions to the EAT-26 were used to identify pathogenic behaviors, including binge eating, self-induced vomiting or purging, use of weight control supplements such as laxatives, diet pills, and diuretics (water pills), excessive exercise to lose or control weight, and loss of 20 pounds (9.072 kg) or more in the last six months. The first four pathogenic behavior questions were answered on a Likert scale of 1–5, and the final question was answered with a yes or no response. To be considered at risk, a participant’s score needed to be greater than 20 and/or meet the criteria for pathogenic behavior risk. Those who scored below 20 with no pathogenic behavior risk were deemed not at risk for ED behaviors [26]. The EAT-26 questionnaire has been validated and used in previous studies with collegiate athletes [27], has a reliability of 0.90 [26], and the reliability for this study was 0.92. A total of 268 participants completed the EAT-26 portion of the study.

#### 2.1.3. Sex-Specific Figural Stimuli Silhouette

The Sex-Specific Figural Stimuli Silhouette (SIL) was used to assess BID between PBI and DBI [28] (Figure 1). The scale consists of nine images depicting body silhouettes increasing in size denoted by increasing numbers of 1 through 9. Each silhouette is anchored to a specific body mass index that are denoted in Figure 1 [28]. Participants were asked to identify which number and corresponding SIL best represented their current PBI and their DBI in their daily clothing, and their PBI and DBI in their uniform (midriff and full-length). Additionally, participants were asked to identify the PBI and DBI of the SILs from the perception of their MP peers, MP parents, and MP coaches. A total of 256 participants completed the SIL to assess PBI and DBI in the various clothing types, while 163 participants completed the SIL to assess PBI and DBI for meta-perceptions.

### 2.2. Procedures

We used a snowball sampling method where a survey link via Qualtrics (Qualtrics, Inc., Provo, UT, USA) was sent through email to All-Star and college coaches and athletic trainers who worked with cheerleading teams. These individuals were asked to forward the survey link to any cheerleader they had access to in both settings. Second, a research team member, in collaboration with Varsity Spirit, attended regional cheerleading competitions to recruit participants. A subdivision of Varsity Spirit, called Varsity University, supplied a booth for the researchers, which allowed for the recruitment of additional participants before or after competitions. This in-person tactic allowed the research team to ensure parental invitation was completed prior to minor cheerleaders participating in the survey. The research team utilized a QR code which allowed potential participants to access the online survey at any time. The first page of the online survey included the invitation/consent letter followed by the demographic survey, the EAT-26, and the Sex-Specific SILs to assess BID. The survey was open for 30 days. A total of 560 surveys were initiated; however, due to the nature of the survey, participants were allowed to skip specific questions based on their comfort in answering. This presented varying sample sizes for the EAT-26 (*n* = 268), SIL for clothing type (*n* = 164), and SIL for meta-perceptions (*n* = 164). The sample sizes are provided within Table 1, Table 2, Table 3 and Table 4. Power was achieved with a minimum of 163 participants for all analyses.

### 2.3. Statistical Analysis

Data were collected and exported from the web-based survey platform to SPSS (SPSS Inc., Version 27, Armonk, NY, USA) for all analyses. Alpha level was set at *p* < 0.05. Using G*Power Statistical software [29] for a repeated measures ANOVA including factors with a small to moderate effect size of 0.3, the power calculation indicated a sample of 20 participants per group for a total of 60, which would have an estimated power of 0.95. Means and standard deviations for age, current weight, highest weight, lowest weight, ideal weight, height, BMI, and EAT-26 subscales were completed for the overall sample. Frequencies and proportions were used to determine the ED type risk and risk of pathogenic behaviors for the overall sample, team type, squad type, and position. Independent samples t-test were conducted to determine differences between age, current weight, highest weight, lowest weight, ideal weight, height, BMI, and team type. Additional independent sample t-tests were used to determine differences in EAT-26 subscale scores across team type and squad type. Chi-square tests of association were conducted to determine differences in ED risk and pathogenic behavior risk across team type, squad type, and position. A one-way ANOVA with Tukey’s post-hoc adjustment was conducted to determine differences in EAT-26 subscale scores across position. One-way within subjects, repeated measures ANOVA models with six values for PBI and DBI were conducted to determine differences in body image perceptions across clothing types (daily clothing, midriff uniform, and full-length uniform), as well as differences in body image across meta-perceptions (peers, parents, coaches). The Greenhouse and Geisser correction were used to correct for violations of sphericity.

## 3. Results

### 3.1. Participant Characteristics

A total of 268 cheerleaders were included in the study: All-Star (*n* = 134), College (*n* = 134); All-girl (*n* = 173), Coed (*n* = 95); Flyer (*n* = 88), Base (*n* = 126), Back spot (*n* = 53). Self-reported age, height, current weight, ideal weight, highest weight, and lowest weight for participants are presented in Table 1. Aside from age, which differed by design (All-Star: 16.0 ± 2.4 vs. college: 19.8 ± 1.3 years; *p* ≤ 0.001), no significant differences were found between collegiate and All-Star cheerleaders for other self-reported variables.

### 3.2. Eating Disorder Risk

Overall, 34.3% (*n* = 92) of participants were identified as being at risk for an ED. When examining the source of ED risk, 4.1% (*n* = 11) were at risk on the EAT-26 only, 17.9% (*n* = 48) were at risk based on pathogenic behaviors only, and 12.3% (*n* = 33) were at risk from both the EAT-26 and pathogenic behaviors. A significant difference was found between ED risk and cheerleading team type (All-Star vs. college; χ21,268 = 5.363, *p* = 0.021), with college cheerleaders being at a higher risk for ED compared to the All-Star team discipline. No significant differences were found between ED risk and squad type (all-girl vs. co-ed; χ2 = 0.011, *p* = 0.917) or cheerleading position (flyer, base, back spot: χ2 = 0.572, *p* = 0.751). The distribution of being at risk for EDs within position groups were flyer: 37.5% (*n* = 33/88), bases: 32.5% (*n* = 41/127), and back spot: 34.0% (*n*= 18/53). All ED data can be found in Table 2.

### 3.3. Eating Attitudes

Descriptive statistics for EAT-26 subscales are presented in Table 3. There were significant differences between dieting subscale and team type (F(1,266) = 4.065, *p* = 0.045), with the college team type reporting higher means than the All-Star team type. There were no significant differences between total EAT-26 and team type (F(1,266) = 3.206, *p* = 0.075), bulimia subscale and team type (F(1,266) = 3.665, *p* = 0.057), or oral control subscale and team type (F(1,266) = 2.664, *p* = 0.104). When examining by squad type, significant differences were found between total EAT-26 and squad type (F(1,266) = 5.698, *p* = 0.018) and between the oral control subscale and squad type (F(1,266) = 9.897, *p* = 0.002), with the co-ed squad reporting higher means for both subscales, respectively. There were no significant differences between dieting subscale (F(1,266) = 2.344, *p* = 0.127) or bulimia subscale (F(1,266) = 2.414, *p* = 0.121) and squad type. When examining eating attitudes across positions, there were no significant differences between total EAT-26 and position (F(1,266) = 0.303, *p* = 0.739), dieting subscale and position (F(1,266) = 0.694, *p* = 0.501), bulimia subscale and position (F(1,266) = 0.105, *p* = 0.901), or oral control subscale and position (F(1,266) = 1.713, *p* = 0.182).

### 3.4. Pathogenic Behaviors

Descriptive statistics for pathogenic behaviors are presented in Table 4. A small but meaningful proportion of cheerleaders was at risk for each pathogenic behavior: 15.3% (*n* = 41) for binge-eating, 11.9% (*n* = 32) for vomiting, 11.9% (*n* = 32) for use of laxatives, diet pills, and diuretics, 5.2% (*n* = 14) for over-exercising, and 5.2% (*n* = 14) for loss of 20 lbs. When examining by team type, significant differences were found between laxative, diet pill, and diuretic use (χ(21,268) = 6.956, *p* = 0.008), with the college cheerleaders having a higher proportion at risk (17.2%, *n* = 23) compared to All-star cheerleaders. There were no significant differences between team type and binge eating (χ(21,268) = 3.549, *p* = 0.060), vomiting (χ(21,268) = 1.411, *p* = 0.235), over exercise (χ(21,268) = 0.301, *p* = 0.583), and the loss of 20 lbs. (χ(21,268) = 0.01, *p* = 0.583). When examining the risk of pathogenic behaviors and squad type, there were no significant differences between binge eating (χ(21,268) = 0.036, *p* = 0.850), vomiting (χ(21,268) = 2.074, *p* = 0.150), laxative, diet pill, and diuretic use (χ(21,268) = 0.067, *p* = 0.796), over exercise (χ(21,268) = 1.367, *p* = 0.242), and loss of 20 lbs. (χ(21,268) = 3.038, *p* = 0.081). When examining the risk of pathogenic behaviors and position, there were no significant differences between binge eating (F(2,266) = 2.377, *p* = 0.095), vomiting (F(2,266) = 0.672, *p* = 0.511), laxative, diet pill, and diuretic use (F(2,266) = 0.635; *p* = 0.531), over-exercising (F(2,266) = 0.432, *p* = 0.650), and loss of 20 lbs (F(2,266) = 0.311, *p* = 0.733).

### 3.5. Body Image

Data for body image variables are presented in Figure 2 and Figure 3. Body image perceptions were significantly different across different clothing types in cheerleaders (F(2.301, 586.879) = 126.784, *p* < 0.0001 η2 = 0.332). Body image perception values presented differences from PBI to DBI with a large effect size, meaning that this sample of cheerleaders felt they wanted to be smaller across all clothing types. Cheerleaders perceived themselves to be the largest when wearing the midriff uniform and perceived themselves to be the smallest in the full-length uniform. Body image perceptions showed statistically significant differences across meta-perceptions (F(3.397, 550.346) = 19.110, *p* < 0.001, η2 = 0.106) with differences between PBI to DBI with a medium effect size, meaning that this sample of cheerleaders wanted to be smaller across all meta-perception levels. Cheerleaders perceived that coaches viewed them to be the largest, while parents viewed them as the smallest.

## 4. Discussion

The purpose of this study was to examine the overall prevalence of ED risk, eating attitudes, and pathogenic behaviors of competitive cheerleaders across team type, squad type, and position. Additionally, we examined the body image perceptions of cheerleaders and various clothing types, as well as the body image perceptions and meta-perceptions of cheerleaders. This study is unique because it is the first to study ED risk among the All-Star cheerleading population, to examine differences within team type, and include a diverse group of positions.

### 4.1. Eating Disorder Risk

Overall, 34.4% of our sample was at risk for EDs. These findings are like previously reported ED risk ranging from 25–42% within aesthetic athletes [13,15,16,30]. Additionally, our results are comparable to ED prevalence rates previously reported in other non-aesthetic sports and physically active populations: 8% of soccer athletes [31], 11% of elite female athletes from various sports [32], 29% of auxiliary units [14], 32% of ROTC cadets [33], and 42% of equestrian riders [34]. Specifically, within the college cheerleading population, ED risk prevalence was previously reported at 33.1%, consistent with our findings [3]. When examining by team type, the All-Star cheerleading population reported 27.6% at risk for EDs compared to 41.0% of college cheerleaders. An explanation for the difference in risk percentages when comparing the two team types comes from the traditional trajectory of All-Star cheerleading participation being a precursor to college participation. All-Star cheerleading requires early sports specialization, in most cases beginning as early as six years old. This allows for most cheerleaders to have over 5–10 years of cheerleading experience by the time they enter the college setting. Additionally, it has been identified that females aged 17–26 years demonstrate a heightened risk for EDs and will experience pressures in the college setting, which may lead them to attempt to change their body shape and appearance [16]. In this sample of youth cheerleaders, ED risk rates and EAT-26 scores support this notion.

Currently, no literature exists that examines ED risk by squad type, however we found no significant differences in ED risk between all-girl and co-ed teams. Aside from the obvious difference in squad type being the inclusion of males, all-girl and co-ed teams’ function similarly and are scored in the same subjective fashion that would present pressure to all participants. It could be expected that females who do participate on a co-ed team may experience additional pressures from being around male cheerleaders, however this was not supported by our findings. Therefore, more research is needed in this area.

When examining ED risk by cheerleading position, our results demonstrate no differences, while previous findings found flyers were at greater odds of experiencing risks for EDs compared to bases and back spots [3]. Over the last decade, All-Star and college teams have increased overall athleticism and performance factors, resulting in all levels and team types requiring an immense amount of athleticism from participants. It is recommended that specific resources for all teams, squads, and positions should focus on proper fueling techniques (i.e., timing of meals and make up of meals) for the demands of cheerleading skills and performances. Governing organizations, such as the USASF and the NCAA, should work to support and create specific resources and training for coaches, gym owners, and program administrators related to body image and disordered eating so the information can be more readily disseminated to the athletes.

Overall, EDs within athletic populations have been linked to long-term health consequences, such as exasperating components of the female athlete triad, a condition which includes low energy availability, menstrual cycle dysfunction, and low bone mineral density [35]. These conditions, coupled with documented results of ED risk ranging from 27–40% with the cheerleading population [3,15], support the long-term goal for this athletic population to have access to healthcare professionals, specifically athletic trainers, on a day-to-day basis. While this is not always feasible for all levels of the cheerleading population, it is recommended that coaches and administrators minimally identify healthcare providers who can be contacted on a case-by-case referral basis who may assist in the identification, monitoring, treatment, and education of any athlete who may exhibit risk factors for EDs.

### 4.2. Pathogenic Behaviors

High rates of pathogenic behaviors that were found in our sample included binge eating, purging, and the use of laxatives, diet pills, or diuretics. Our results are consistent with other aesthetic sports, such as equestrian, auxiliary units (majorettes and color guards), as well as other cheerleaders, which demonstrated binge-eating ranging from 11–24%, and purging from 9–11% [3,14,34]. However, our findings for the use of laxatives, diet pills, or diuretics was lower than the reported rates of 15–19% in previous literature [3,14,34]. This difference may be explained by the collaboration between the NCAA and Varsity Spirit, Inc which sought to establish risk management guidelines for the sport [3]. However, our results found a significant difference between the use of this behavior in the All-Star and college participants, with college cheerleaders reporting significantly higher rates. This may be due to the obvious age difference between these two squad types, with most college cheerleaders being adults over the age of 18 years. This age difference may allow for college cheerleaders to experience the freedom to purchase and use laxatives, diet pills, or diuretics without the oversight of a parent or guardian. Additionally, college cheerleaders are faced with lifestyle changes when transitioning to college, which often include weight gain [36]. College cheerleaders may idealize the quick results that are seen in weight loss when using laxatives, diet pills, and diuretics. Added education should be provided to the college cheerleading population about the health consequences related to the engagement in these pathogenic behaviors.

### 4.3. Body Image

When examining the BID of participants in relation to their self-perception in various clothing types, a similar trend was found compared to the previous literature [3,14,34]. Participants chose silhouettes for their DBI that were smaller than their PBI for all clothing choices. Consistent with the previous cheerleading literature [3], participants in the present sample reported lowest DBIs for the midriff uniform, which is the most revealing of the clothing types. This finding supports the cultural norm of females idealizing a smaller body shape and size [9]. Additionally, this finding uncovers the potential for self-monitoring within the population of female cheerleaders, which can morph into high degrees of BID, leading to self-objectification [9,22] and increase the risk of ED behaviors to achieve or maintain the ideal body size [16,37]. Within both cheerleading team types, midriff uniforms were frequently utilized. Within the All-Star category specifically, midriff uniforms are worn by senior level teams, which creates an appeal for younger athletes to strive to be placed on higher levels to be awarded the opportunity to wear this uniform style. This, coupled with the early sports specialization that occurs within the All-Star discipline, an environment may be created, which increases self-objectification and negative eating attitudes and behaviors over a long-term period. The USASF has implemented guidelines for the use of the midriff uniform and determined appropriate ages that can perform wearing this uniform type. However, there is a need for further examination into whether this uniform type provides any additional benefit to the overall cheerleading performance for all team and squad types. Individuals who are tasked with choosing uniform styles should fully understand the risks to body image, a documented precursor of EDs, that using midriff uniforms creates for the athletes and allow for athletes’ opinions prior to style selection.

Another unique focus of this study was the meta-perception BIDs from the perspective of coaches, parents, and peers, social agents in the cheerleading environment linked as correlates of disordered eating [4,17,38,39]. Within our sample, cheerleaders reported a smaller silhouette for the DBI for all three meta-perceptions, indicating cheerleaders felt that peers, parents, and coaches wanted them to be smaller than their perceived status. The largest meaningful discrepancy in our study occurred between PBI and DBI and MP coaches, which is consistent with a previous study [4]. Coaches have been identified as being the most prominent influence on athletes at the interpersonal level [40]. With this influence, it is important for coaches to understand the impact of their behaviors and actions on athletes’ mental health and overall body image. It has been documented that coaches may act more favorably towards athletes who exhibit a body type that aligns with their personal desires, and is deemed to be a more appropriate body size for the sport of cheerleading [41]. This favoritism can greatly affect a cheerleader who may perceive themselves as not aligning with those sport-specific body ideals. Athletes who may be taller, heavier, or have a higher BMI have reported perceiving that their coaches are less likely to engage in positive coaching behavior [41]. Therefore, coaches should be aware of the potential impact they have on athletes and be cautious of commenting on weight and body size towards any athlete.

## 5. Limitations and Future Research

While this was the first study to include the All-Star cheerleading population, the following limitations should be considered. The EAT-26 was used to assess eating attitudes and behaviors. While a validated tool, the measure alone should not be used to diagnose individuals, therefore we cannot conclude that the participants in this study who were classified at risk in fact had a clinical ED. Additionally, the EAT-26 is a self-reported measure; therefore, the authors must assume all participants responded honestly and accurately. Finally, cheerleading has many participants worldwide, therefore the results of our study cannot be generalized to the entire population. While the sampling method allowed for a large sample size within this data set, the majority of participants were from the Midwest and Southeast regions of the United States and should not be considered as a completely representative sample. Future research should include a larger sample size from a variety of geographical regions. Additionally, ED risk should be evaluated in male cheerleaders.

## 6. Conclusions

The findings of this study indicate cheerleaders, both in the All-Star and college setting, are at risk for EDs and BID. Over one-fourth of All-Star cheerleaders and over one-third of college cheerleaders were identified as experiencing attitudes and behaviors associated with EDs. These findings highlight the need for education towards the ED risk which these athletes face, and increased education surrounding the topic of overall health and well-being for cheerleaders of all ages is warranted. Moreover, our findings indicate that the midriff uniform and coaches have the largest impact on the BID of female athletes. Coaches should consider this impact when choosing uniform types, and should further evaluate this clothing option to determine its use in the future of both the All-Star and college settings. It is recommended the USASF and NCAA consider incorporating adequate training for coaches and gym owners so they understand the impact they have on their athletes, as well to advocate for more of the use of healthcare professionals, such as athletic trainers, for medical oversight.

## Figures and Tables

**Figure 1 ijerph-19-02196-f001:**
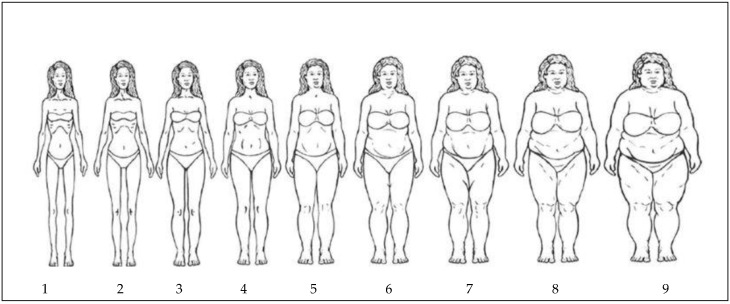
Sex-Specific Figural Stimuli Silhouettes.

**Figure 2 ijerph-19-02196-f002:**
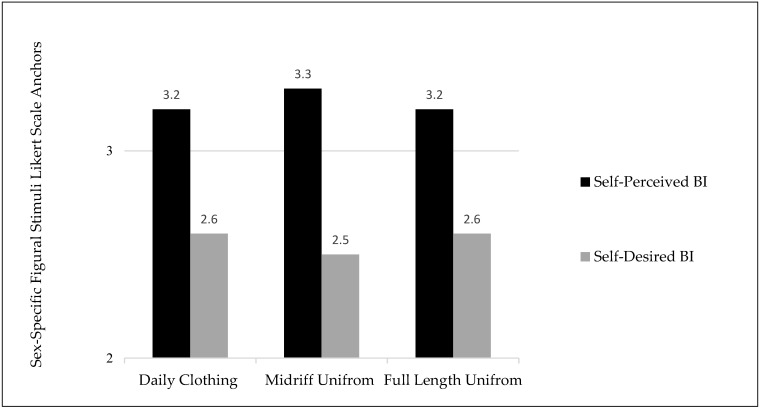
Mean body image values for perceived and desired body image across clothing types (daily clothing, midriff uniform, and full-length uniform) using Sex-Specific Figural Stimuli Likert scale anchors.

**Figure 3 ijerph-19-02196-f003:**
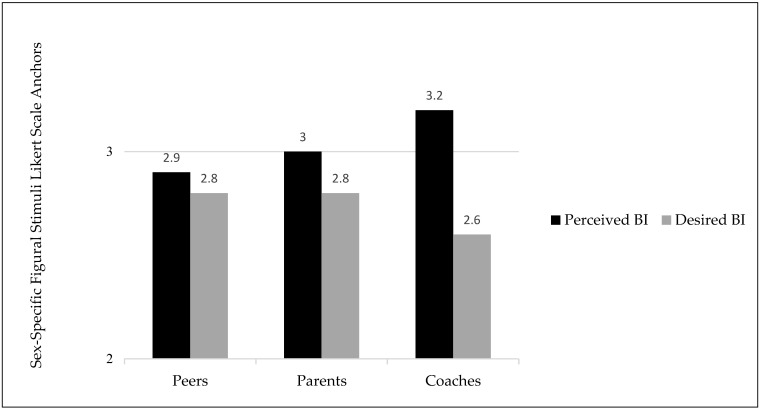
Mean body image values for perceived and desired body image across meta-perceptions (peers, parents, and coaches) using Sex-Specific Figural Stimuli Likert scale anchors.

**Table 1 ijerph-19-02196-t001:** Descriptive statistics presented as mean (standard deviation) for self-reported age, weight, height, and body mass index.

	Females			
	All (N = 268)	All-Star (N = 134)	College (N = 134)	
	Mean ± SD	Mean ± SD	Mean ± SD	*p*-Value
Age (years)	17.9 ± 2.7	16.0 ± 2.4	19.8 ± 1.3	≤0.001
Weight (kg)				
Current	58.7 ± 11.7	57.9 ± 11.8	59.5 ± 11.6	0.378
Highest	61.0 ± 13.4	59.5 ± 13.9	62.6 ± 12.9	0.391
Lowest	51.0 ± 14.5	47.9 ± 16.6	55.9 ± 9.8	0.033
Ideal	54.5 ± 9.4	53.1 ± 9.0	54.1 ± 9.8	0.330
Current-Ideal	4.2 ± 6.0	4.8 ± 7.5	3.6 ± 3.9	0.240
Height (cm)	161.5 ± 14.0	163.3 ± 18.1	159.6 ± 8.1	0.891
BMI (kg/m^2^)	16.1 ± 3.12	16.5 ± 3.4	15.8 ± 2.9	0.198

*p*-value for mean differences between All-Star and collegiate cheerleaders.

**Table 2 ijerph-19-02196-t002:** Proportions of participants classified as at risk for ED for the entire sample and by cheer team type, squad type, and position.

	Overall at Risk % (*n*)	EAT-26 % (*n*)	Pathogenic Behavior % (*n*)	Both % (*n*)
All Participants (*n* = 268)	34.3 (92)	4.1 (11)	17.9 (48)	12.3 (33)
**Team Type**				
All-Star (*n* = 134)	27.6 (37) *	4.5 (6)	15.7 (21)	7.5 (10)
College (*n* = 134)	41.0 (55) *	3.7 (5)	20.1 (27)	17.2 (23)
**Squad Type**				
All-girl (*n* = 173)	34.1 (59)	4.6 (8)	20.2 (35)	9.2 (16)
Co-ed (*n* = 95)	34.7 (33)	3.2 (3)	13.7 (13)	17.9 (17)
**Position**				
Flyers (*n* = 88)	37.5 (33)	5.7 (5)	18.2 (16)	13.6 (12)
Bases (*n* = 127)	32.5 (41)	2.4 (3)	17.5 (22)	12.7 (16)
Back Spot (*n* = 53)	34.0 (18)	5.7 (3)	18.9 (10)	9.4 (5)

* Significant (*p* < 0.05) difference of proportion at risk of ED between groups.

**Table 3 ijerph-19-02196-t003:** Descriptive statistics presented as mean ± standard deviation for total EAT-26 score and EAT-26 subscale scores.

	Total EAT-26	Dieting	Bulimia	Oral Control
	Mean ± SD	Mean ± SD	Mean ± SD	Mean ± SD
All Participants (*n* = 268)	10.6 ± 10.7	6.4 ±7.6	2.1 ± 2.4	2.2 ±2.7
**Team Type**				
All-Star (*n* = 134)	10.3 ± 0.9	7.1 ± 0.6 *	2.1 ± 0.2	2.9 ± 0.3
College (*n* = 134)	11.1 ± 1.0	8.0 ± 0.7 *	2.7 ± 0.2	2.5 ± 0.2
**Squad Type**				
All-girl (*n* = 173)	9.6 ± 0.7 **	7.1 ± 0.5	2.2 ± 0.2	2.2 ± 0.2
Coed (*n* = 95)	12.4 ± 1.3 **	8.5 ± 0.9	2.6 ± 0.3	3.4 ± 0.3
**Position**				
Flyers (*n* = 88)	11.1 ± 9.5	7.2 ± 7.2	2.1 ± 2.2	1.9 ± 2.3
Bases (*n* = 127)	10.7 ±11.4	6.1 ± 7.7	2.1 ± 2.6	2.5 ± 3.0
Back Spot (*n* = 53)	9.7 ± 11.1	5.8 ± 8.1	1.9 ± 2.3	1.9 ± 2.6

* Significant (*p* < 0.05) difference for dieting subscale and team type. ** Significant (*p* < 0.05) difference between total EAT-26 score and squad type.

**Table 4 ijerph-19-02196-t004:** Proportion of participants classified as at risk for pathogenic behaviors for the entire sample and by cheer team type, squad type, and position.

	Binge Eating % (*n*)	Vomiting % (*n*)	Laxatives, Diet Pills, Diuretics % (*n*)	Excessive Exercise % (*n*)	Lost 20 lbs. (9.07 kg) % (*n*)
All Participants (*n* = 268)	15.3 (41)	11.9 (32)	11.9 (32)	5.2 (14)	5.2 (14)
**Team Type**					
All-Star (*n* = 134)	12.7 (17)	8.2 (11)	6.7 (9)	4.5 (6)	6.0 (8)
College (*n* = 134)	17.0 (24)	15.7 (21)	17.2 (23) *	6.0 (8)	4.5 (6)
**Squad Type**					
All-girl (*n* = 173)	15.6 (27)	9.8 (17)	11.6 (20)	4.0 (7)	3.5 (6)
Coed (*n* = 95)	14.7 (14)	15.8 (15)	12.6 (12)	7.4 (7)	8.4 (8)
**Position**					
Flyers (*n* = 88)	20.5 (18)	9.1 (8)	13.6 (12)	6.8 (6)	4.5 (4)
Bases (*n* = 127)	10.3 (13)	14.3 (18)	12.7 (16)	4.0 (5)	6.3 (8)
Back Spot (*n* = 53)	18.9 (10)	11.3 (6)	7.5 (4)	5.7 (3)	3.8 (2)

* Significant (*p* < 0.05) difference between laxatives, diet pills, and diuretics, and team type.

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
