# Peer review of "Investigation of Eating Disorder Risk and Body Image Dissatisfaction among Female Competitive Cheerleaders"

_ijerph, 2022, doi:10.3390/ijerph19042196_

Round 1
Reviewer 1 Report
I enjoyed reviewing this article. Well-written and nicely presented. The work addressed the core point of the cheerleader segment of sports which often is not investigated up to the scale. Therefore, this work would certainly get a buzz in this arena. However, I would like to mention some minor corrections before accepting the article for publication. Please see them below-
L(ine) 1- Title: I suggest replacing ‘Examination’ with ‘Investigation’. Based on the study design, investigation suits better to describe this research article.
L30-31- What could be the reason/ mechanism behind this outcome?
L42- ……….ages (5-18). Ref? I guess it is not your own statement.
L58- …..(5-14). Please use en-dash instead of hyphen. Follow the same in other several places.
L60- ……youth population. Not clear. What is the check-point of ‘focus’?
L60-61- ……..outdated(ref?), low sample (Ref?) and other sports (Ref?). We along with the future readers would like to see the specific reference.
L62- 13-33%. Use en-dash please.
L62-64- Very strong/ loud voice. There is some literature available, I guess. Therefore, suggest adjusting the statement- like- Only a few research………
L69- Replace Body Image Dis with BID
L107- -ages 12 to 25 years old- Why? What is the rationale for choosing this band? Provide ground, please.
L130- Online survey. A questionnaire needs to be provided as an appendix. So that the reviewer and the future readers could see the base (repeatability).
L134-135- Res team…………kgs. You may remove this statement (common conversion in practice).
L137- Again, where is the list of Qs to accomplish EAT26. Put it in the appendix please.
L209-210- The age gap seems 4 years (16-19.8), needs to be highlighted and discussed in the current investigation as older ones might contribute very differently than the young ones.
L282-285- Fig 2 and 3 could be represented in tables/ even in the text also.
L423-424- Do you think the culture and weather might also have an impact of clothing of CLs.
Author Response
Reviewer 1
I enjoyed reviewing this article. Well-written and nicely presented. The work addressed the core point of the cheerleader segment of sports which often is not investigated up to the scale. Therefore, this work would certainly get a buzz in this arena. However, I would like to mention some minor corrections before accepting the article for publication. Please see them below-
L(ine) 1- Title: I suggest replacing ‘Examination’ with ‘Investigation’. Based on the study design, investigation suits better to describe this research article.
Thank you for your kind overall feedback and suggestion for the Title change. The title has been changed to reflect the Investigation suggestion.
L30-31- What could be the reason/ mechanism behind this outcome?
Thank you for the suggestion for content to be added to the abstract. However based on the word count criteria set by the journal, there is no additional room for content to be added to this section of the manuscript.
L42- ……….ages (5-18). Ref? I guess it is not your own statement.
There is no reference to be added since this is just a set range to delineate between all star cheerleading and college cheerleading.
L58- …..(5-14). Please use en-dash instead of hyphen. Follow the same in other several places.
En-dash was used.
L60- ……youth population. Not clear. What is the check-point of ‘focus’?
Additional clarification was added to outline that the “youth” population we are targeting is specifically the under 18 year olds.
L60-61- ……..outdated(ref?), low sample (Ref?) and other sports (Ref?). We along with the future readers would like to see the specific reference.
References were added throughout this sentence.
L62- 13-33%. Use en-dash please.
En-dash was added between the numbers
L62-64- Very strong/ loud voice. There is some literature available, I guess. Therefore, suggest adjusting the statement- like- Only a few research………
No literature was replaced with minimal literature.
L69- Replace Body Image Dis with BID
There is no “Body image Dis” appearing in the section the reviewer outlined? No changes were implemented to line 69.
L107- -ages 12 to 25 years old- Why? What is the rationale for choosing this band? Provide ground, please.
More rationale was added to the sentence to help it become more clear why this age range was selected.
L130- Online survey. A questionnaire needs to be provided as an appendix. So that the reviewer and the future readers could see the base (repeatability).
Due to the research teams ongoing research study in this area, we have decided to not include the survey used as an appendix.
L134-135- Res team…………kgs. You may remove this statement (common conversion in practice).
Sentence was removed.
L137- Again, where is the list of Qs to accomplish EAT26. Put it in the appendix please.
The EAT-26 and additional questions are freely available for anyone to access. The authorship team has decided not to add it as an appendix.
L209-210- The age gap seems 4 years (16-19.8), needs to be highlighted and discussed in the current investigation as older ones might contribute very differently than the young ones.
Based on the inclusion criteria for age of all star and age of college cheerleaders, this age gap was a known difference prior to data collection.
L282-285- Fig 2 and 3 could be represented in tables/ even in the text also.
The information that is represented in Fig 2 and 3 is discussed throughout the section 3.5 Body Image.
L423-424- Do you think the culture and weather might also have an impact of clothing of CLs.
Thank you for this thoughtful suggestion. While anecdotally the coorseponding author does feel that culture has an impact on the athletes wanting to wear the midriff uniform, no data were collected related to the topics of culture or weather.

Reviewer 2 Report
It is interesting study in which authors investigated the status of eating Disorder Risk and Body Image Dissatis-2 faction among Female Competitive Cheerleaders. Due to smaller sample size and specific nature of sampling technique, the conclusion should be treated with caution. The following issues should be addressed further before the publication.
- Due to specific sampling, sample could be real representative one. More discussion about the nature of this sample should be included.
- Authors said in the part of 2.2 procedures that “A total of 560 surveys were initiated; however, there were varying sample sizes for the EAT-26, SIL for clothing type, and SIL for meta-perceptions. The sample sizes are provided within Tables 2-5. Power was achieved with a minimum of 163 participants for all analyses.”. it sound confusing. How many participants they wanted? For each content, how many participants respond to? About power analysis, what indicators were considered for sample size estimation? This statement is suggested to be clarified.
- The statement on repeated measures ANOVA models with six values for PBI and DBI is also confusing to me. Where is this result?
- Limitation on sample itself should be discussed further.
- It would be better if it is possible to assess the association between ED and body image.
Author Response
Reviewer 2
It is interesting study in which authors investigated the status of eating Disorder Risk and Body Image Dissatisaction among Female Competitive Cheerleaders. Due to smaller sample size and specific nature of sampling technique, the conclusion should be treated with caution. The following issues should be addressed further before the publication.
- Due to specific sampling, sample could be real representative one. More discussion about the nature of this sample should be included.
We have added a statement the in limitations section of the manuscript to outline why this sample is not a true representative one of the cheerleading population.
- Authors said in the part of 2.2 procedures that “A total of 560 surveys were initiated; however, there were varying sample sizes for the EAT-26, SIL for clothing type, and SIL for meta-perceptions. The sample sizes are provided within Tables 2-5. Power was achieved with a minimum of 163 participants for all analyses.”. it sound confusing. How many participants they wanted? For each content, how many participants respond to? About power analysis, what indicators were considered for sample size estimation? This statement is suggested to be clarified.
Sample sizes were added for the EAT-26, and SIL measures.
- The statement on repeated measures ANOVA models with six values for PBI and DBI is also confusing to me. Where is this result?
The results for the ANOVA analysis is presented in the Body Image section as well as figure 2 and 3.
- Limitation on sample itself should be discussed further.
We have added a statement the in limitations section of the manuscript to outline why this sample is not a true representative one of the cheerleading population.
- It would be better if it is possible to assess the association between ED and body image.
Thank you for the suggestion. The authors will take this into consideration for future studies.

Reviewer 3 Report
Adolescence is a special period in child development. Thus, often the process of biological maturity progresses faster than a child's maturation. It can lead to different emotional and behavioral disorders. Cheerleading participation can induce or intensify potential abnormalities.
The authors assessed the risk of body image dissatisfaction and eating disorders in a group of 268 female cheerleaders. Team type, squad type, position, clothing type and meta-perceptions of peer, parents and coaches in the examined group were analyzed. The study is very interesting and novel. Even though the data are based on judgment call, the article provides a lot of important information about female competitive cheerleaders.
1. The manuscript familiarizes us with certain risks of certain disorders. The authors provided a detailed analysis of behavioral and eating disorders depending on the position on the cheerleading team. For example, according to the presented data, the group wearing midriff uniforms should be periodically examined by a psychologist.
2. A weak point is that this is not an objective method of testing (judgment call), since the method used was the snowball sampling method, where not all patients were included.
3. Also, the tests should be performed on a larger examined group and analysis should be separated (All-star and college, separately) in the future. It would be nice for the authors to include how many hours a day/ week each group practiced and duration of sport experience.
Author Response
Reviewer 3
Adolescence is a special period in child development. Thus, often the process of biological maturity progresses faster than a child's maturation. It can lead to different emotional and behavioral disorders. Cheerleading participation can induce or intensify potential abnormalities.
The authors assessed the risk of body image dissatisfaction and eating disorders in a group of 268 female cheerleaders. Team type, squad type, position, clothing type and meta-perceptions of peer, parents and coaches in the examined group were analyzed. The study is very interesting and novel. Even though the data are based on judgment call, the article provides a lot of important information about female competitive cheerleaders.
- The manuscript familiarizes us with certain risks of certain disorders. The authors provided a detailed analysis of behavioral and eating disorders depending on the position on the cheerleading team. For example, according to the presented data, the group wearing midriff uniforms should be periodically examined by a psychologist.
- A weak point is that this is not an objective method of testing (judgment call), since the method used was the snowball sampling method, where not all patients were included.
- Also, the tests should be performed on a larger examined group and analysis should be separated (All-star and college, separately) in the future. It would be nice for the authors to include how many hours a day/ week each group practiced and duration of sport experience
Thank you so much for the thoughtful and kind feedback. The authors appreciate your time in reviewing this manuscript and will take your suggestions into account for future studies.

Round 2
Reviewer 2 Report
Thanks for addressing my comments, and I have not any concerns.